# Comparison and benchmark of name-to-gender inference services

Lucía Santamaría[1,*] and Helena Mihaljević[2,*]

[1] Amazon Development Center, Berlin, Germany
[2] University of Applied Sciences, Berlin, Germany
[*] These authors contributed equally to this work.

## ABSTRACT

The increased interest in analyzing and explaining gender inequalities in tech, media, and academia highlights the need for accurate inference methods to predict a person's gender from their name. Several such services exist that provide access to large databases of names, often enriched with information from social media profiles, culture-specific rules, and insights from sociolinguistics. We compare and benchmark five name-to-gender inference services by applying them to the classification of a test data set consisting of 7,076 manually labeled names. The compiled names are analyzed and characterized according to their geographical and cultural origin. We define a series of performance metrics to quantify various types of classification errors, and define a parameter tuning procedure to search for optimal values of the services' free parameters. Finally, we perform benchmarks of all services under study regarding several scenarios where a particular metric is to be optimized.

# INTRODUCTION

Quantitative measurements and large-scale analyses of social phenomena in relation to gender are gaining significance as tools to uncover areas of gender bias and inequality, and to ultimately foster women's inclusion and advancement. Algorithms that can infer the gender category from other features have thereby opened up new opportunities to enhance data previously lacking such information. Examples include social media profiles, GitHub contributors, and authors of scientific publications, the analysis of which regarding gender has led to a better understanding of women's situation in domains such as tech (*Vasilescu, Serebrenik & Filkov, 2015*), media (*Matias, Szalavitz & Zuckerman, 2017*; *Macharia et al., 2015*), and academic publishing (*Larivière et al., 2013a*; *West et al., 2013*; *Mihaljević-Brandt, Santamaría & Tullney, 2016*; *Bonham & Stefan, 2017b*).

Particularly in the latter case of bibliometric studies, the most reliable piece of information available for guessing a gender is the name string of the author. The standard approach for name-to-gender inference is based upon querying large (and often openly available) name repositories, such as censuses, administration records, and universal or country-specific birth lists. Occasionally, results are refined with name-forming rules for specific cultures or ethnicities. In the attempt to identify the gender of as many names as

Corresponding author
Lucía Santamaría,
lucia.santamaria@ymail.com

possible, often a multi-step combination of database queries, insights from sociolinguistics, and manual corrections is performed. This leads to non-transparent processes for gender inference that might be troublesome to reproduce and almost impossible to test, compare, and transfer.

Recently, the plethora of self-labeled data arising from social media has been successfully leveraged to improve the accuracy and specificity of methods based on querying compiled lists of names. This input has given rise to a handful of free and paid web services that infer gender from name strings. They usually gather data from manifold sources and profit from a greater degree of diversity, notably in regard to the origin of names, thus becoming a good choice for analyses outside of a national context. Access to such services is typically granted through APIs, turning gender inference on large corpora into a fast, reliable, and cost-effective process. Using them in large-scale analyses is tempting due to their accuracy and availability; nonetheless, some caveats may apply, as the underlying data sources are frequently closed and thus not necessarily reliable nor verifiable.

It is perhaps no surprise that, with few exceptions, the majority of research that uses name-based gender assignment does not evaluate the chosen method nor justifies the decision to use particular data sources and inference methodologies. Furthermore, only a handful of studies have attempted to compare different approaches and provide solid groundwork for the choice of a given tool. We intend to fill this gap by presenting a comprehensive benchmark and comparison of several available gender inference services applied to the identification of names stemming from the academic publishing community.

We evaluate web services *Gender API*, *genderize.io*, *NameAPI*, *NamSor* and Python package *gender-guesser*, five popular and commonly used methods for the problem at hand. All services are able to handle an international set of names and are thus singularly valuable for bibliographic analyses. After describing the services broken down by several decision-critical properties, such as data sources, accessibility, and cost, we test each of them on a manually labeled data set consisting of 7,076 names, which we make publicly available to interested researchers working on this topic (*Mihaljević & Santamaría, 2018*).

Several metrics of relevance for the classification problem are defined, distinguishing between *misclassifications*, i.e., assignments of the wrong gender to a name, and cases for which it is not possible to predict a gender, denominated *non-classifications*. To optimize for the metrics we perform repeated, cross-validated, randomized searches over the free parameters of the services. For Gender API, genderize.io, and NamSor we report error rates under 15% for inaccuracies including non-classifications, under the constraint that the misclassification error amounts to a maximum of 5%. The three services also achieve less than 2% error for misclassifications when imposing that at least 75% of all names be assigned a gender. Gender API is in general the best performer in our benchmarks, followed by NamSor.

The cultural context of a name is known to have an impact on gender inference; to assess its importance we have used NamSor's origin API to predict the most likely origin of the names and split our analysis with respect to this facet. As expected, the less confident predictions occur for names of Asian origin. We quantify the effect of the names'

 

provenience on the results of the surveyed gender assignment services; overall, Gender API outperforms all others for Asian names as well.

To the best of our knowledge, this is the most comprehensive comparison of gender assignment algorithms for the field of bibliometric analyses performed to date. The results of our analysis are meant to be the basis for further claims of gender assignment accuracy in future studies on the topic.

## Related work

Bibliometric studies based on systematic assignment of gender with the purpose of analyzing specific aspects of the academic landscape have been conducted for at least a decade. *Mozaffarian & Jamali (2008)* studied the productivity of Iranian women in science by analyzing over 2,500 publications by Iranian authors from 2003, taken from WoS. Gender inference was done manually, resorting to an internal list of Iranian academics and to Internet searches. While not scalable, this approach showed that a high degree of familiarity with names from a particular country greatly aids the gender inference task. The article reported lower productivity of Iranian female authors with respect to their male counterparts. With a much broader scope, *Frietsch et al. (2009)* considered patent applications from 14 European countries between 1993 and 2001 and publications in eight scientific areas from 1996 to 2006, extracted from Scopus. Comprehensive and country-specific lists of names collected, post-processed, and tested by *Naldi et al. (2005)* were applied to assign a gender to inventors and authors. The analyzed data set comprised almost 2,500,000 inventors and 500,000 authors from over 150,000 publications, after rejecting over 60% of names that could not be assigned a definite gender. Findings included stark national differences in female contributions, with central European countries being the most prone to exhibit a wide gender gap.

Recently, various studies have focused on large-scale, scalable bibliometric analyses of academic publications in relation to gender. *West et al. (2013)* analyzed over 1,500,000 publications from JSTOR, a digital library corpus in science and humanities. They could assign a gender to 73% of authors with full first names by using data from the US Social Security Administration records. Their analysis showed that authorships by women are not evenly distributed over the dimensions time, field, coauthorship, and authorship position. A similar approach was followed by *Larivière et al. (2013a)*, who resorted to both universal and country-specific sources such as the US and Quebec censuses, WikiName portal, Wikipedia, and various Internet sites to assign a gender to all articles between 2008 and 2012 indexed in WoS. This resulted in more than 5,000,000 articles and over 27,000,000 authorship instances, 86% of which could be assigned a gender, provided a full first name was available. They reported significant gender disparities, such as fewer citations for articles with women in dominant author positions, as well as underrepresentation of women as first authors. Analogous findings, but restricted to mathematics, were reported by *Mihaljević-Brandt, Santamaría & Tullney (2016)*, who analyzed over 2,000,000 publications from bibliographic service zbMATH between 1970 and 2014. Using the names list from *Michael (2007)*, they could assign a gender to 61% of all authorship instances. Most recently, *Holman, Stuart-Fox & Hauser (2018)* estimated the gender gap in STEMM

disciplines with help of the PubMed and arXiv databases. Thirty-six million authors with publications over the last 15 years were assigned a gender using genderize.io, which is one of the services that we evaluate. Their results confirm previously reported high variations of the gender gap across countries. Furthermore, according to their data model, gender parity won't be reached during this century in various fields, including mathematics, physics, and computer science.

It is worth mentioning that all four above-mentioned studies performed some kind of validation of their gender inference methods, yet there is room for improvement regarding assessment of manual gender labels, data size, or reproducibility. For instance, *Holman, Stuart-Fox & Hauser (2018)* estimate their gender misclassification rate to be 0.3% based on a collection of 372 manually labeled author names via Web searches. Considering the expected name variance, an extrapolation of the error estimate to the entire data set does not seem reliable.

Despite its importance for estimating the error rate on gender assignments, only a few studies are devoted to comparing different gender inference tools. *Vanetta (2014)* tested and compared four gender assignment methods (*beauvoir*, *Sexmachine*, *genderPredictor*, and genderize.io) on a control group provided by a government office and consisting of over 400 first names. No claims were made as to which one of the services was better, arguing that different purposes may pose different requirements to the services.

*Karimi et al. (2016)* used the test set from *Larivière et al. (2013a)* to evaluate precision, recall and accuracy of several methods, including data from various censuses, genderize.io, face recognition algorithm *Face++*, and two novel approaches consisting of mixing the predictions of the latter two. They reported improved accuracy of the mixed methods, noting that the quality of the assignments depended on country and was worse for non-Western regions. The brevity of the paper prevents an extended discussion regarding the definition of the quality metrics, particularly in the handling of the predicted unknowns. At any rate, face recognition techniques clearly hold potential, albeit they must be used with caution in view of their likely intrinsic bias, e.g., towards darker-skinned females (*Buolamwini, 2017*). Similarly, equating country of residence with regional origin of a name does not seem to be a well suited assumption, given that academics often move internationally. *Holman, Stuart-Fox & Hauser (2018)* also query genderize.io with the country of affiliation, potentially incurring the same bias.

A more extensive comparison is that of *Wais (2016)*, who revisited the methods and results of *Larivière et al. (2013a)* and *West et al. (2013)*, with the additional introduction of the R package *genderizeR* based on the genderize.io service. To compare the three approaches, a common test data set was evaluated that contained 2,641 authorships of 2,000 articles manually labeled using Internet searches. The results were compared in terms of the metrics described in 'Performance metrics'. The method based on genderize.io outperformed the others, at least with respect to metrics that focus on retrieving a gender for as many names as possible. The percentage of non-classifications was consequently the lowest. While genderize.io and genderizeR seem to offer the best performance for gender prediction, the author points out the bias towards English names.

**Table 1  Comparison table showing relevant features for the gender inference services under study.** Note that although Gender API does provide a specific API end point for handling surnames, our results employ the version that does not make use of them.

|  | Gender API | gender-guesser | genderize.io | NameAPI | NamSor |
|---|---|---|---|---|---|
| Database size (January 2018) | 1,877,787 | 45,376 | 216,286 | 510,000 | 1,300,000 |
| Regular data updates | yes | no | yes | yes | yes |
| Handles unstructured full name strings | yes | no | no | yes | no |
| Handles surnames | yes | no | no | yes | yes |
| Handles non-Latin alphabets | partially | no | partially | yes | yes |
| Implicit geo-localization | no | no | no | yes | yes |
| Assignment type | probabilistic | binary | probabilistic | probabilistic | probabilistic |
| Free parameters | accuracy, samples | – | probability, count | confidence | scale |
| Open source | no | yes | no | no | no |
| API | yes | no | yes | yes | yes |
| Monthly free requests | 500 | unlimited | 30,000 | 10,000 | 1,000 |
| Monthly subscription cost (100,000 requests/month) | 79 € | Free | 7 € | 150 € | 80 € |
| Provider | Gender-API.com | Israel Saeta Pérez | Casper Strømgren | Optimaize GmbH | NamSor SAS |

Recently, *Bonham & Stefan (2017b)* have published an analysis of female underrepresentation in the field of computational biology, for which they performed a preliminary comparison of three gender assignment tools (*Bonham & Stefan, 2017a*). The methods tested were the Python package *GenderDetector* and web APIs genderize.io and Gender API. After inferring the gender of 1,000 names they ultimately chose Gender API for its superior coverage. They did not compute further metrics to validate their election.

Lastly, a crucial discussion of the benefits as well as ethical concerns when using gender inference methods was posed in *Matias (2014)*, which presented a multitude of examples of meaningful projects that have applied such tools to various data sources. Links to evaluations and comparisons of name-based gender inference services, including the Open Gender Tracker, which the author co-developed, were provided as well.

## METHODS

### Overview of surveyed services

We compare five different services for inferring gender from name strings that are among the methods most frequently employed to perform gender assignments in the field of bibliometric studies. Several of them are broadly used by organizations and companies in the private sector as well. Table 1 showcases key characteristics; below we briefly describe each of them.

#### *Gender API*

Gender API (https://gender-api.com/), a gender inference service launched in 2014, offers a standard first name search with capability to handle double names. Furthermore, the API allows queries with a full name, which is internally split into first and last. The service currently supports 178 countries, although it cannot geo-localize a full name per se. Its API

accepts extra parameters for localized queries though, namely country code, IP address, and browser locale. The response contains gender assignments *male*, *female*, or *unknown*, plus confidence parameters *samples* and *accuracy*. The former is the number of database records matching the request, while the latter determines the reliability of the assignment. The service is built upon a combination of data from multiple sources, partially from publicly available governmental records combined with data crawled from social networks. Each name has to be verified by different sources to be incorporated. The service is overall reliable, with its cloud-based infrastructure providing an availability of 99.9%.

### Python-package gender-guesser

Python package gender-guesser (https://pypi.python.org/pypi/gender-guesser/) implements a wrapper around the names data described in *Michael (2007)*, which was made available in *Michael (2008)*[1]. The data set comprises over 45,000 names with gender assignments *unknown* (name not found), *andy* (androgynous), *male*, *female*, *mostly_male*, or *mostly_female*. Additionally, the approximate frequency of each name per country is provided, and the gender request can be made with an extra location parameter for better accuracy. The dictionary of names was published a decade ago and has not been updated since, which limits the usefulness of both the package and its underlying data source. On the other hand, the gender assignment of this collection is presumed to be of high quality, with manual checks by native speakers of various countries.

### genderize.io

Online service genderize.io (https://genderize.io/), created in August 2013, attempts to infer the gender of a first name. The response is either *male*, *female*, or *None*, plus two additional confidence parameters, *count* and *probability*, representing the number of data entries used to calculate the response and the proportion of names with the gender returned in the response. The underlying data is collected from social networks across 79 countries and 89 languages. Although the service does not geo-localize names automatically, it does accept two optional parameters, *location_id* and *language_id*, for more qualified guesses. The providers do not state the sources employed, hence the reliability of the data is difficult to assess. An API and extensions to various languages are available. There are no guarantees about uptime; the service might not be reliable enough for use in critical applications.

### NameAPI

NameAPI (https://www.nameapi.org/) is a free and paid service platform to work with names. It provides functionality in the form of web services to do name parsing, genderizing, matching, formatting, and others. For our benchmark we have concentrated on the genderizing service only. Its underlying data sources are dictionaries with all parts of names extracted from census publications, birth lists, and telephone books from over 55 countries. Original spellings in non-Latin scripts (including transcriptions and transliterations) are also recorded. The gender response can be *MALE*, *FEMALE*, *NEUTRAL*, *UNKNOWN*, or *INDETERMINABLE*, weighted by the *confidence* parameter. The service is able to infer the most likely origin of the name, thus allowing to apply language-specific rules and to geo-localize names whose gender depend on the culture. The service aims to achieve

[1]Other Python wrappers of the same data are *Genderator* and *SexMachine*, but they are mostly unsupported and not Python3-compatible.

high uptime, with 99.999% availability. NameAPI provides an API accessible either from a free or a paid account, with credits that can be purchased on a one-time or monthly subscription basis.

### NamSor

NamSor (http://www.namsor.com/) is a classifier of personal names by gender, country of origin, or ethnicity. The service is able to recognize the linguistic or cultural origin of names, thus allowing it to correctly infer the gender from the first and last name in cases that can be male or female depending on their provenience. NamSor claims to cover all languages, alphabets, countries, and regions. The underlying data consists of 1.3 million unique given names extracted from baby name statistics per country, plus sociolinguistic information (morphology, language, ethnicity, etc.) to extract semantics, which allows it to predict gender for complex cases. The NamSor gender API returns *male*, *female*, or *unknown* values, plus a parameter *scale* ranging from −1 to +1 to reflect the certainty that a name is male or female, respectively. A basic API for structured names is available for free, whereas the Freemium version accepts unstructured strings and offers higher performance and precision. NamSor's origin API recognizes the likely cultural provenience of personal names in any alphabet, returning a primary and an alternative potential country of origin, as well as a score to qualify the trustworthiness of the assignment. It is based on a proprietary onomastics model which uses linguistic and cultural information.

## Assemblage of test data

We have gathered, revised, and combined human-annotated author-gender data sets used in various bibliometric studies to date, which we describe below.

### zbMATH

Randomly selected authors from the bibliographical records of the mathematical publications service zbMATH (https://zbmath.org/), sampled in 2014 ignoring names that contained only initials. These authors were manually labeled as 'female', 'male' or 'unknown' by *Mihaljević-Brandt, Santamaría & Tullney (2016)* using Internet queries to obtain gender information. More precisely, the concrete person behind an author's name was identified by gathering author profile information from zbMATH and other bibliographic databases. Then, university websites, Wikipedia articles, and similar online sources were searched for gender-indicating titles (Mr, Mrs, etc.) and personal pronouns corresponding to the according person. The *zbmath* data set consists of 400 names (291 male, 58 female, 51 unknown).

### genderizeR

Sample data sets from the genderizeR package (https://github.com/kalimu/genderizeR), *authorships* and *titles* that correspond to records of articles of *biographical-items* or *items-about-individual* type, respectively, from all fields of study, published from 1945 to 2014 and drawn from WoS. As described in *Wais (2016)*, the names in both data sets were manually coded as 'female', 'male', or 'unknown' based on results from Internet queries using the authors' full names, affiliations, biographies, mentions in the press, and photos.

A name was deemed 'unknown' if the coder was not certain enough based on the found information. We have applied a series of preprocessing steps to this data, namely removal of duplicates and names containing only initials. For the 'titles' data set we have used a naive named-entity extractor based on Python's package *nltk*'s POS tagger to identify names in the articles' titles. Generally, NER tasks are better solved with more potent packages, such as Stanford's *CoreNLP*. In this case though, the small size of the data set allowed for a manual check of all extracted names to guarantee correctness. After revising both sources, *genderize_r_titles* and *genderize_r_authors*, the data set consists of 1,037 names (740 male, 145 female, 152 unknown).

### PubMed

Data set from *Filardo et al. (2016)*, built by querying the six journals with highest JCR impact factor in 2012 in the category *Medicine, general & internal* for original research articles between 1994 and 2014. Incidentally, this data has also been used to tune the gender assignment methods of *Bonham & Stefan (2017b)*. *Filardo et al. (2016)* determined the gender of the first author of each of the articles as 'female', 'male', or 'unknown' by first inspecting the forename. If this was judged to be insufficient to assign a gender, the authors searched institutional websites, social media accounts, photographs, and biographical paragraphs to gather more information. We further removed duplicates and records with empty first names or initials only. The *pubmed* data set consists of 1,952 names (1,209 male, 714 female, 29 unknown).

### WoS

Data set produced for the validation study reported in *Larivière et al. (2013b)* that informed the findings of *Larivière et al. (2013a)*, consisting of records from the WoS database covering all publications from 2008 to 2012 included in Science Citation Index Expanded, the Social Sciences Citation Index, and the Arts and Humanities Citation Index. From each of the five categories 'initials', 'unknown', 'unisex', 'male', and 'female', 1,000 names were randomly sampled and associated with a specific country, institution, and, in some cases, an email address. This information was used by *Larivière et al. (2013a)* to locate biographical information and, based on that, manually assign a gender. From this data set of 5,000 names we have further removed records from the 'initials' subset and duplicates with respect to the previous data sets. The final *wos* data set consists of 3,687 names (1,571 male, 1,051 female, 1,065 unknown).

After concatenating all data sets we ran a sanity check consisting of finding names that had been consistently misclassified by all gender inference services, and performed manual corrections to amend incorrectly assigned labels. In total, we double-checked 74 author names and manually changed the gender label of 46 of them by searching preferably for personal pronouns, gender-indicating titles, and, ultimately, photos in university websites, professional social media sites, conference web pages, and similar sources. All undertaken preprocessing steps can be found in *Mihaljević & Santamaría (2018)*.

Our final data set consists of 7,076 names (3,811 male, 1,968 female, 1,297 unknown), split into three components: first, middle, and last name. About 13% of them contain a middle name, and the number of unique combinations of first and middle name in the data

| Table 2 | Examples of the geographical origins of names as inferred by NamSor's origin API. | | | | | |
|---|---|---|---|---|---|---|
| Full_name | Gender | Source | Country | Top_region | Sub_region | Score |
| maria bortolini | f | wos | Italy | Europe | Southern Europe | 2.925873 |
| liew woei kang | m | pubmed | China | Asia | Eastern Asia | 2.638786 |
| sirin yasar | f | wos | Turkey | Asia | Western Asia | 3.357177 |

set is 3,956. The most common male names are 'John', 'David', and 'Michael'; the most frequent female ones are 'Susan', 'Christine', 'Laura', and 'Anne'. We point out that we use 'female' and 'male' as categories solely because the evaluated services operate under the gender binary paradigm; our concerns to this approach are spelled out in the discussion. To the best of our knowledge this collection is the largest sample of labeled gender data corresponding to authors of academic publications to date. In order to promote further research on the topic, we make our data set available to interested researchers (*Mihaljević & Santamaría, 2018*).

## Origin of the names

The assembled data set described above does not include any information regarding the origin or geographic provenience of the persons' names. It is well known that the cultural context is an important aspect affecting the reliability of gender inference methods. We thus seek to evaluate the geographical and cultural diversity of our data set as well as to measure the impact of the names' origins on the performance of the surveyed services. As described above, NamSor's origin API is able to produce an anthroponomical classification of a personal name by allocating an onomastic class, i.e., a country, subregion, and region, to it. The inferred origin is reported in conjunction with a *score* that calibrates its reliability. According to NamSor's internal evaluations, for names from Europe, Asia, and Africa the classifications with *score* > 0 are trustworthy. Note that the USA is not considered an onomastic class on its own but rather a melting pot of other 'cultural origins' such as Ireland or Germany (*Carsenat, 2014*)[2]. Similarly, names from other parts of the Americas are considered to be of (Southern) European descent. Table 2 shows a few examples of the anthroponomical classifications produced by NamSor's origin API.

[2]NamSor provides a diaspora API to address onomastic classification for countries with historically high immigration rates like the USA.

   We have applied NamSor's origin API to our collection of 7,076 names and are able to assign a provenience to 97% (6,866) of them that return a *score* above 0, and that we keep for further analysis. The service was queried in May 2018. NamSor estimates 4,228 names (61%) to be of European origin, 2,304 (34%) of Asian, and 334 (5%) of African provenience. We have split the analysis by the different data sources; results are displayed in Fig. 1. The wos collection contains approximately 45% Asian and 50% European names; given the fact that wos is larger than any of the other data subsets, this ensures a satisfactory representation of Asian names in the whole test data set. For the other data sources, names of European origin clearly predominate, especially in the genderize_r subsets. In short, our data set shows a majority of European and Asian names; understandably for a sample coming from scientific publishing, the proportion of African names is small.

   A more fine-grained geographical analysis focusing on countries, rather than regions, reveals that the names in our test data originate from 113 countries; however, just 16 of

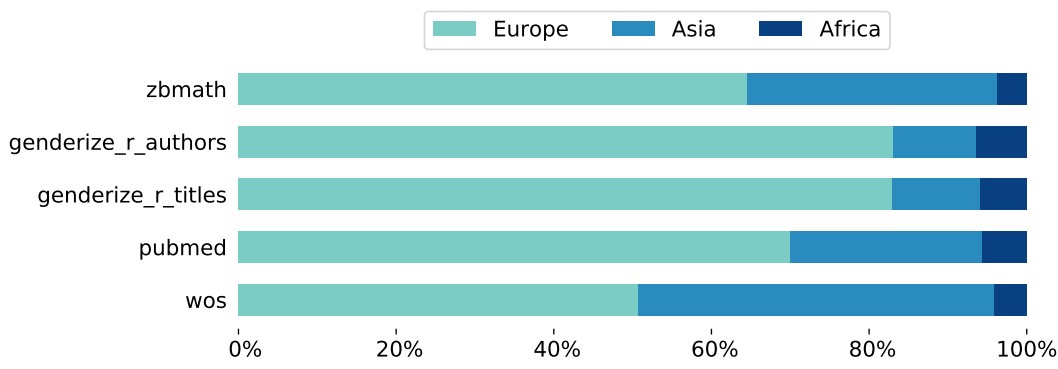

**Figure 1** **Geographical region of origin of the personal names from our test data set as inferred by NamSor's origin API.** The colored bars show percentages split by data sources. The genderize_r data sets are the most Eurocentric, whereas the wos collection is more balanced towards Asian names. African names amount to at most 6% per data source, which reflects the shortage of scholarly authors from that region.

them already cover 75% of the whole data set. The most frequent country of origin is the UK, followed by Germany, China, and Ireland. Splitting the analysis among the five data sources confirms that both genderize_r datasets are very Eurocentric: they have almost no Asian countries among the top 10 most frequent ones. They also show the lowest diversity, with the top three countries (UK, Germany, and Ireland) covering 50% of all names and the top 10 covering 75%. On the contrary, the highest variability appears in the smallest data set zbmath, where the top three countries (Germany, Japan, and the UK) contain 26% of all names and the top 10 cover 60%. The larger wos data set exhibits the best representation of Asian origins: China, Japan, and Korea appear in positions 1st, 3rd, and 6th in terms of frequency. Full figures and statistics can be found in our dedicated Jupyter notebook in *Mihaljević & Santamaría (2018)*.

The analysis of cultural and geographical provenience of the personal names contained in our data set brings us to the conclusion that our collection is reasonably diverse and shows an acceptable variability with respect to origin, with the aforementioned caveats about African names.

## Retrieval of gender assignments

We have performed systematic queries to the services under study to obtain an inferred gender for all records in our test data set. All queries were performed in mid December 2017. Depending on their peculiarities, we sent different requests to the various APIs. Concretely, genderize.io and gender-guesser do not handle last names, therefore we queried using first and middle names only. When both a first and a middle name were present we tried their combinations (e.g., 'Jae Il', alternatively 'Jae-Il'). If no response was obtained, only the first name was used (e.g., 'Jae'). The free NamSor API requires the name to be structured as split forename(s) and last name. Gender API offers two distinct end points, one for forename(s) only and another capable of splitting unstructured full name strings. We evaluated both methods in our benchmark and found that the former performs notably better, thus we

report results for that variant. NameAPI accepts unstructured strings per se. We tested it using the full name string and the forename(s) only, for better comparison with the name splitting mechanism of Gender API. However, in this case querying with the full name string achieves significantly better results, hence we report the performance of NameAPI for this variant.

## Performance metrics

Name-based gender inference can be considered as a classification problem, and as such there exist plenty of metrics to measure the performance of a classifier. The choice of performance indicators needs to be suitable for the problem at hand and is usually a non-trivial task in the development of data-based algorithms. For further background readings related to classification and learning algorithms in general, such as training and testing, cross-validation, typical error estimates and statistical significance testing, see e.g., *Hastie, Tibshirani & Friedman (2009)* and *Japkowicz & Shah (2014)*.

The names in our evaluation data set have been manually labeled as 'female', 'male', or 'unknown'. Recall that those labeled as 'unknown' refer to individuals for whom it was not possible to find sufficient gender-related information online. Therefore the class 'unknown' is rather a heterogeneous label applied to people with either very common names or those that avoid providing much personal information online. In particular, it is not a 'gender class' in any suitable sense, and cannot be included appropriately in quantitative evaluations using performance metrics. In what follows, we will not make use of the items with manual label 'unknown' for any of our calculations and results, working instead with the 5,779 names in our data set which possess a defined gender label. (For a discussion of ethical considerations of name-based gender inference methods and the methodological shortcomings of such approaches, see the 'Discussion'.) On the other hand, the services evaluated here do return, along with the responses 'female' and 'male', at least a label 'unknown' for the unidentified cases. Hence, in terms of the true labels we are dealing with a binary classification problem, while the predictions contain one or more extra output classes.

This makes it difficult to pose name-based inference as a standard classification problem and to utilize commonly used metrics such as precision and recall. In our case it makes sense to work instead with metrics derived from the confusion matrix defined as follows:

|  | predicted class | | |
|---|---|---|---|
| | male | female | unknown |
| **male** | $m_m$ | $m_f$ | $m_u$ |
| **female** | $f_m$ | $f_f$ | $f_u$ |

(true class)

Let us introduce the following nomenclature for the components of the confusion matrix, which in general we will refer to as *assignments*: elements in the diagonal ($m_m$ and $f_f$) are the *correct classifications*, while elements outside it ($m_f$ and $f_m$) are thus *misclassifications*. The sum of both can be simply referred to as *classifications*. Consequently, elements $m_u$ and $f_u$ represent *non-classifications*, since the algorithm fails at predicting one of the classes

'male' or 'female'. All mistakes, both misclassifications and non-classifications, are included under the term *inaccuracies*. Based on the confusion matrix, *Wais (2016)* introduced four performance metrics:

$$\text{errorCoded} = \frac{f_m + m_f + m_u + f_u}{m_m + f_m + m_f + f_f + m_u + f_u},$$

$$\text{errorCodedWithoutNA} = \frac{f_m + m_f}{m_m + f_m + m_f + f_f},$$

$$\text{naCoded} = \frac{m_u + f_u}{m_m + f_m + m_f + f_f + m_u + f_u},$$

$$\text{errorGenderBias} = \frac{m_f - f_m}{m_m + f_m + m_f + f_f}.$$

The errors above are to be interpreted as follows: *errorCoded* treats a non-classification as a regular error and penalizes it in the same way as a misclassification, therefore it encodes the fraction of inaccuracies over the total number of assignments; *errorCodedWithoutNA* measures the share of misclassifications over the total number of classifications while ignoring non-classifications; *naCoded* computes the proportion of non-classifications over the total number of assignments; *errorGenderBias* estimates the direction of the bias in gender prediction, indicating whether there are more females misclassified as male, or vice versa. If positive, then the estimated number of women is higher than in the real data.

Depending on the concrete usage of an algorithm, these metrics can be suitable or not. For instance, if not being able to assign a gender to a large number of names is acceptable while high prediction accuracy is essential, *errorCodedWithoutNA* should be minimized. For most purposes, however, it is desirable to infer the gender for as many names as possible without treating non-classifications as a regular error. For this purpose we have defined two extensions of the metrics above.

Let $w \in [0, 1]$. We define the *weightedError* as

$$\text{weightedError}_w = \frac{f_m + m_f + w * (m_u + f_u)}{m_m + f_m + m_f + f_f + w * (m_u + f_u)}.$$

For $w = 0$ the *weightedError* equals *errorCodedWithoutNA* and for $w = 1$ we recover *errorCoded* exactly. For $0 < w < 1$ we define a metric which penalizes misclassifications more than non-classifications. To clarify this further, consider the following examples of confusion matrices:

|        | male | female | unknown |
|--------|------|--------|---------|
| male   | 11   | 2      | 6       |
| female | 1    | 7      | 4       |

$C_1 = $

|        | male | female | unknown |
|--------|------|--------|---------|
| male   | 11   | 6      | 2       |
| female | 4    | 7      | 1       |

$C_2 = $

For both confusion matrices the fraction of inaccuracies over all assignments is the same (and equals 0.419), while $C_1$ exhibits a smaller proportion of misclassifications, given that the number of non-classifications is larger than in $C_2$:

$$\text{errorCoded}_{0.2}(C_1) = \frac{1+2+0.2*(6+4)}{11+1+2+7+0.2*(6+4)} = 0.217,$$

$$\text{errorCoded}_{0.2}(C_2) = \frac{4+6+0.2*(2+1)}{11+4+6+7+0.2*(2+1)} = 0.371.$$

Another even simpler possibility to penalize non-classifications without giving them the same importance as to misclassifications is to minimize a metric such as *errorCodedWithoutNA*, which ignores the class 'unknown', while enforcing a constraint on *naCoded*, i.e., the rate of non-classifications. Indeed any two metrics can be combined in this way. However, a disadvantage of this approach is that, for certain constraint values, there is no solution in the parameter space of a given gender inference model. Thus, it makes sense to consider the distribution of error values depending on the tuning parameters before setting a definite constraint value. In our benchmarks, if the constraint cannot be satisfied on a training set, we have fixed the test set error to 1, which is the maximum achievable value.

## RESULTS

Prior to parameter tuning and benchmarking we have performed sample requests to all services in order to test potentially problematic cases, such as double names and diacritics. All services under study are sensitive towards accents and return different responses when e.g., applied to the names 'José María' and 'José Maria'. However, NamSor and NameAPI show less sensitivity than Gender API and genderize.io. For instance, NameAPI returns the same value of the free parameter for both 'Jose'/'José' and 'Maria'/'María', respectively, but makes a difference when queried with a double name.

The handling of double names is actually not completely transparent for most of the services. In the cases of 'Mary-Jane' or 'Jane-Mary', Gender API returns a *count* value resulting of adding those for 'Mary' and 'Jane'. This pattern persists when concatenating further names of the same gender, e.g., as in 'Mary-Jane-Sarah'. Yet when name parts are joined with empty space instead of hyphen, the count values are not added, which shows that a different logic is being applied. The response of genderize.io also depends on the character connecting the name parts. This indicates a low level of semantical preprocessing

of the data sources for both services. We have found examples of similar behavior in NamSor, while NameAPI seems to be less susceptible to this kind of artifacts.

As pointed out in *Wais (2016)*, names used in social network profiles may contain arbitrary words or characters. This 'bogus data' is not filtered out of the underlying data base of genderize.io, resulting in stop words like 'with' or 'I' having a gender assigned to them. The same is true to a similar extent for Gender API, while NameAPI and NamSor show a higher level of data curation. The package gender-guesser contains a priori only vetted names.

Both NameAPI and NamSor make use of the surname in order to provide a more accurate gender guess for names which depend significantly on the cultural context. We tested this on a few examples such as 'Andrea Schmidt' vs. 'Andrea Bocelli' or 'Rosario González' vs. 'Rosario Giordano'; the two services responded with the expected gender categories, showing a correct identification of the surname with a particular country of origin. Gender API and genderize.io do assign a gender to names like 'Andrea' or 'Rosario', but with lower accuracy values. When queried with unisex first names such as 'Mika', 'Addison', 'Ash', or 'Dakota', NameAPI returns 'neutral' or 'unknown'. NamSor, Gender API, and genderize.io interpret some of these names as gender neutral by assigning a value of their free parameter close to 0, while gender-guesser also treats names as ambiguous via the qualifier *mostly*.

## Parameter tuning

All methods under study with exception of gender-guesser return, in addition to the gender, one or two numerical parameters to estimate the quality of the inference. Figure 2 shows the distribution of these free parameters when using NameAPI, NamSor, Gender API, and genderize.io to assign genders to all names in our test data set. For gender-guesser (not displayed) we have created one such parameter by setting it to 0.75 for responses 'mostly_female' or 'mostly_male' and 1 for 'female' or 'male'. Figure 2A suggests that NameAPI's parameter *confidence* exhibits a bimodal distribution that peaks at around 0.90, with a secondary maximum at 1 (absolute certainty on the gender assignment). All names reach a *confidence* of at least 0.70. Figure 2B indicates that NamSor assigns a gender with absolute certainty to most of the names in the data set, although some outliers are spread out at smaller values of the *scale* parameter. Figures 2C and 2D show the two-dimensional parameter spaces of Gender API and genderize.io: one parameter encodes the number of appearances of a name (denominated *samples* and *count*, respectively), and the other shows the confidence on its gender assignment (named *accuracy* and *probability*). Most names fall in the bottom right region of high confidence and low number of counts.

In general, there is no mathematical model that can be trained in the classical sense of machine learning. Instead, for every service, we have 'trained' (or 'tuned') the algorithms by trying out randomly sampled parameter values. Assuming that a gender classification might not be reliable under some given threshold for the confidence indicators, we have searched for those parameter values that minimize a certain error. A particular instantiation of a random grid of sampled parameters used for each service (except gender-guesser) is displayed as black dots in Fig. 2 as well.

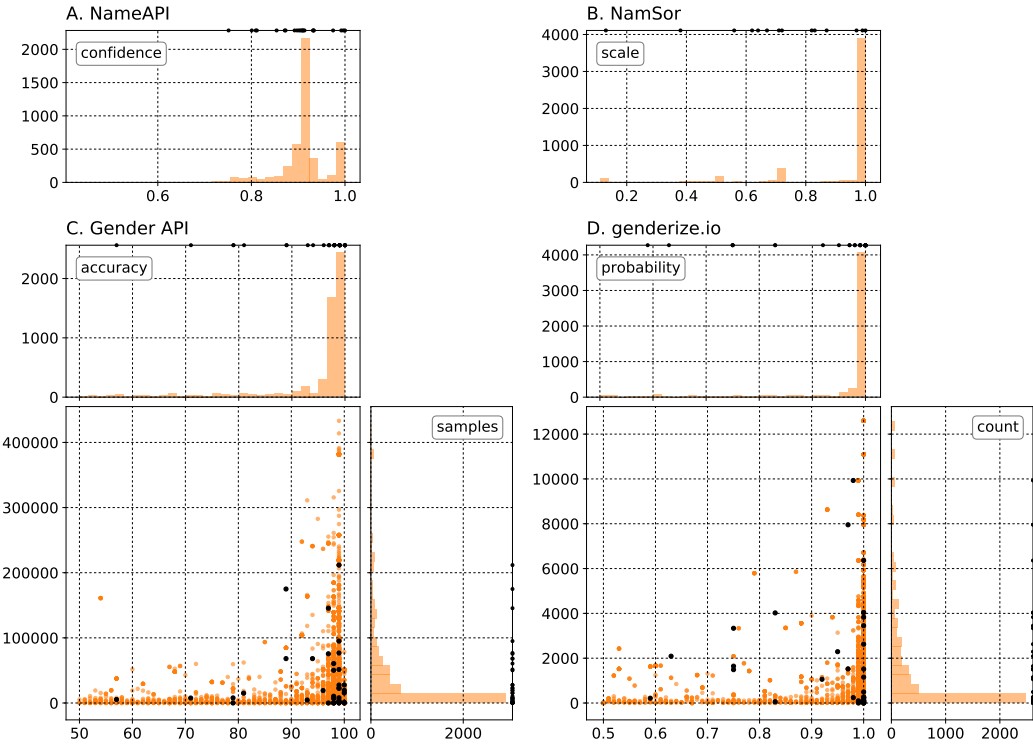

**Figure 2** **Distribution of free parameters after querying the gender inference services with all f/m names from our test data set.** (A) and (B) return one parameter, while (C) and (D) return two. In black, a particular instantiation of the grid of parameters per service used to perform parameter tuning is shown.

## Benchmark setting

We define a series of benchmarks to compare the methods under study. To begin with, we compute performance metrics using the default gender responses for all services, i.e., with no error minimization through parameter tuning (Benchmark 1). Next, we introduce further Benchmarks 2, 3, and 4, each of which concentrates on a particular scenario defined by conditions on the metrics to be optimized.

All performed benchmarks are evaluated two-fold: (a) on the entire data set, and (b) differentiating between the five data sources as described in 'Assemblage of test data'. The latter case is particularly relevant e.g., for researchers in scientometrics looking for the most suitable gender inference algorithm for a particular data source such as PubMed, whereas the former is most appropriate when analyzing different data collections, as e.g., in *Holman, Stuart-Fox & Hauser (2018)*. To better illustrate the role of the names' origin, Benchmark 1b is additionally broken down by geographical regions first and then by Asian subregions, which turn out to be the most challenging.

For Benchmarks 2a, 3a, and 4a we run 10 trials of 10-fold cross-validation. In each, we randomly select (at most) 40 parameter values per service for training, and record the average error on the test sets with those parameters that minimized the respective error on

the training sets. The tuning parameters as well as the training/test set splits are randomly selected with fixed seed numbers, thus ensuring reproducibility.

In Benchmarks 2b, 3b, and 4b we perform one trial of 10-fold cross-validation per service and data set, and record the average error on the test sets. Since the individual data sets differ significantly in size, we have used at most 20 parameters for the smaller data sets zbmath, genderize_r_authors and genderize_r_titles. For the larger data sets pubmed and wos, we have allowed at most 30 and 35 tuning parameters, respectively.

Finally, we apply several tests to assess the statistical significance of the observed differences. In the b-versions of our benchmarks, we first apply the non-parametric Friedman test and then suitable post-hoc tests. We refrain from using ANOVA, which is the usually recommended parametric alternative to evaluate multiple classifiers on multiple data sets, since the homogeneous covariance assumption is not satisfied across all services and data sets. In each benchmark, we compare the two best performers using suitable parametric or non-parametric tests for two classifiers. For more details on statistical significance tests suited for classification algorithms, see *Demšar (2006)* and *Japkowicz & Shah (2014)*[Chapter 6].

The code for the benchmark evaluations can be found in *Mihaljević & Santamaría (2018)*.

## Benchmark 1: Default responses

First, we use the gender inference services employing their default responses, i.e., considering all 'female' and 'male' gender attributions regardless of the value of the confidence parameters.

### Benchmark 1a: entire data set

We report the resulting figures for correct classifications, misclassifications, and non-classifications per service on the entire data set in Table 3, whereas Table 4 shows values of the various quality metrics. Gender API exhibits the lowest fraction of inaccuracies, at 7.9%. It also achieves the smallest proportion of non-classified names, a mere 3%. For both metrics, NamSor is the next best performer, closely followed by genderize.io. Note that the databases of Gender API and NamSor are about one order of magnitude larger than those of gender-guesser and genderize.io, therefore it is not surprising that they achieve a larger ratio of classified names. Incidentally, NameAPI incurs a comparatively high non-classification error, despite its relatively extensive database.

Regarding the metrics that ignore predicted 'unknowns' altogether, Python package gender-guesser achieves the best results, with only 2.6% of misclassifications, followed by NameAPI. This means that, when considering only the proper classifications in the confusion matrix, these services minimize the number of elements outside the diagonal. In other words, they produce the least number of misclassifications when ignoring the non-classifications. This is indicative of a high-quality data curation when incorporating names into the database. Regarding the error in gender bias, the worst offenders are genderize.io and Gender API, although in reverse directions; while the former wrongly identifies more men as women, the latter does the opposite, which means that results

**Table 3 Confusion matrices for all services using their default responses without parameter tuning.**

|  | $m_{pred}$ | $f_{pred}$ | $u_{pred}$ |
|---|---|---|---|
| **(a) Gender API** | | | |
| m | 3,573 | 110 | 128 |
| f | 172 | 1,750 | 46 |
| **(b) gender-guesser** | | | |
| m | 2,964 | 66 | 781 |
| f | 56 | 1,530 | 382 |
| **(c) genderize.io** | | | |
| m | 3,210 | 189 | 412 |
| f | 73 | 1,744 | 151 |
| **(d) NameAPI** | | | |
| m | 3,126 | 93 | 592 |
| f | 75 | 1,616 | 277 |
| **(e) NamSor** | | | |
| m | 3,354 | 132 | 325 |
| f | 94 | 1,684 | 190 |

**Table 4 Benchmark 1a: performance metrics for all services with their default gender assignments on the entire data set.** The *weightedError* is computed with $w = 0.2$.

|  | errorCoded | errorCodedWithoutNA | errorGenderBias | naCoded | weightedError |
|---|---|---|---|---|---|
| Gender API | 0.0789 | 0.0503 | −0.0111 | 0.0301 | 0.0562 |
| gender-guesser | 0.2224 | 0.0264 | 0.0022 | 0.2012 | 0.0731 |
| genderize.io | 0.1428 | 0.0502 | 0.0222 | 0.0974 | 0.0703 |
| NameAPI | 0.1794 | 0.0342 | 0.0037 | 0.1504 | 0.0672 |
| NamSor | 0.1282 | 0.0429 | 0.0072 | 0.0891 | 0.0613 |

obtained with Gender API may underestimate the amount of females. Finally, measured in terms of the weighted error with $w = 0.2$, all services perform quite similarly, with Gender API producing the lowest error.

It is perhaps necessary to point out that, despite the high accuracy level of various gender predictions according to the services' responses, some persons indeed remain misclassified. We can provide several examples extracted from our analysis: 'Dana' is thought to be female with 91% *accuracy* by Gender API, 0.98 *confidence* by NameAPI, and 0.92 *probability* by genderize.io, while NamSor more conservatively sets its *scale* to 0.16. In fact, our test data set includes a male 'Dana'. Similarly 'Michal', the name of a female researcher, is classified as male with 97% *accuracy* by Gender API, 0.93 *confidence* by NameAPI, -0.91 *scale* by NamSor, and 0.75 *probability* by genderize.io. Ultimately it all comes down to internal heuristics and the way each service weights the counts for a particular name from their multiple data sources. Thus it is unavoidable to end up with a number of misclassified names, and this should be taken into account when making absolute claims about the validity of results based on these services.

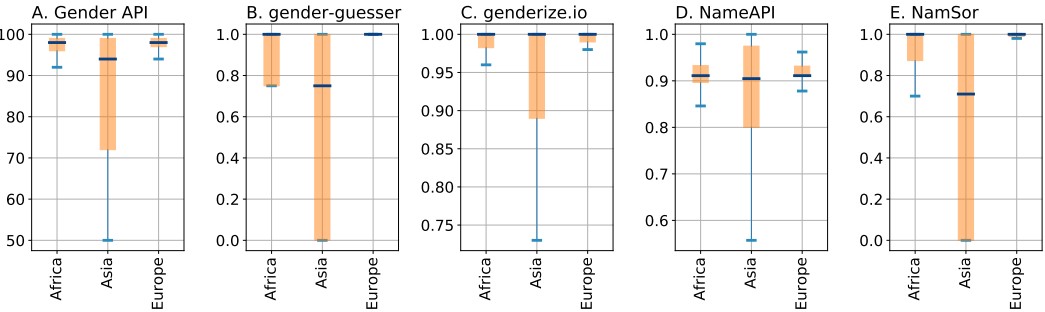

**Figure 3** **Boxplots depicting quartiles for the confidence parameters of the gender inference services, split by geographical regions Africa, Asia, and Europe as returned by NamSor's origin API.** Panels (A), (B), (C), (D), and (E) display parameters *accuracy* of Gender API, self-constructed confidence of gender-guesser, *probability* of genderize.io, *confidence* of NameAPI and *scale* of NamSor, respectively. The bottom and top of the colored boxes mark the first and third quartiles of the distribution; the line in the middle of the boxes indicates the median; the ends of the whiskers correspond to the lowest and highest data points within 1.5 interquartile range.

### Benchmark 1b: analysis by name origin and data source

Next, we investigate the impact of the names' origin on the performance of the services under evaluation. As described above, all services return a confidence parameter indicating the trustworthiness of the classification; recall that for Python package gender-guesser we have created one such parameter by setting it to 0.75 for responses 'mostly_female' or 'mostly_male' and 1 for 'female' or 'male'. We investigate the confidence parameters for different geographical origins of the test names. The boxplots in Fig. 3 show statistics from the quartiles of the parameters' distributions, split by the top regions predicted by NamSor's origin API.

Note that all services produce responses that are discernibly dependent on the names' origin: the most confident gender predictions are for names of European origin, while Asian names generally lead to a lower median and a higher deviation. The service NameAPI displayed in Fig. 3D stands out insofar as the medians of confidence values are lower than those of the other services, indicating a different kind of normalization. This is in accordance with the bimodal parameter distribution peaking at around 0.90 which is depicted in Fig. 2A. There is also little difference between the median values for all three geographical regions, suggesting that NameAPI's *confidence* parameter is not as useful to discriminate among easy versus complex cases. It is also worth noting in Fig. 3C that service genderize.io assigns gender to Asian names with significantly higher confidence values than the other services: the median value for Asian names is surprisingly almost as high as for European names.

The lower confidence in gender assignments for Asian names reported by almost all services suggests focusing on that case. As shown in Fig. 4, which displays results broken down by Asian subregions, names from Eastern and Southeastern Asia yield significantly smaller values than other Asian regions. This is to be expected, in particular due to Chinese names for which the difficulty of inferring a gender of Latin transcriptions is well known.

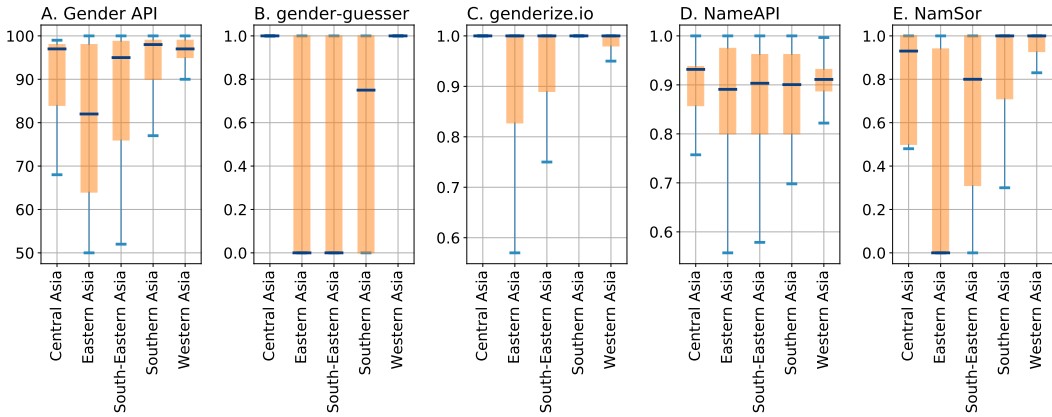

**Figure 4** Boxplots depicting quartiles for the confidence parameters of the gender inference services for Asian subregions as returned by NamSor's origin API, with boxplot settings as in **Fig. 3**.

**Table 5** Benchmark 1b, name origin: performance of all services with their default gender assignments in terms of the metrics *errorCoded* and *errorCodedWithoutNA*, broken down by name origin. Values are rounded to four decimal figures.

| | errorCoded | | | errorCodedWithoutNA | | |
|---|---|---|---|---|---|---|
| | **Africa** | **Asia** | **Europe** | **Africa** | **Asia** | **Europe** |
| Gender API | 0.0538 | 0.1759 | 0.0281 | 0.0469 | 0.112 | 0.0213 |
| gender-guesser | 0.2437 | 0.5171 | 0.0752 | 0.0365 | 0.0641 | 0.0147 |
| genderize.io | 0.1039 | 0.3282 | 0.0507 | 0.053 | 0.1206 | 0.0218 |
| NameAPI | 0.1505 | 0.3772 | 0.0807 | 0.0405 | 0.0897 | 0.0136 |
| NamSor | 0.0645 | 0.3459 | 0.0273 | 0.044 | 0.0903 | 0.0211 |

Nonetheless, while NamSor and gender-guesser have almost no confidence in the gender inference of East Asian names, Gender API shows a notably high median value. NameAPI and genderize.io again exhibit similar medians for all subregions, confirming that the values of their confidence parameters are decoupled from the names' origins, and thus from the complexity of the assignment. This fact makes us doubt that NameAPI's *confidence* and genderize.io's *probability* parameters are sufficiently significant.

Table 5 quantifies errors incurred by the different services depending on the names' origin. Gender API achieves the best results for *errorCoded*, however its performance is strongly affected by the names' origin, being one order of magnitude worse for Asian (18% inaccuracies) than for European names (3%). NamSor performs similarly for European (3%) and African names (7%), but is considerably worse for Asian ones (35%). Regarding the fraction of misclassifications (*errorCodedWithoutNA*) we note that gender-guesser, with its small but highly accurate database, achieves a mere 1.5% error in classifying European names, while for Asian names the figure increases to 6%. Generally speaking, we conclude that all services clearly show the challenging nature of inferring gender for Asian names.

Next let us consider errors in gender inference depending on the source of the names as displayed in Table 6. For both errors all services perform much worse on the wos data subset

**Table 6** Benchmark 1b, data source: Performance of all services with their default gender assignments in terms of the metrics *errorCoded* and *errorCodedWithoutNA*, broken down by data source. Values are rounded to four decimal figures.

| | errorCoded | | | | | errorCodedWithoutNA | | | | |
|---|---|---|---|---|---|---|---|---|---|---|
| | zbmath | genderize_r_ authors | genderize_r_ titles | pubmed | wos | zbmath | genderize_r_ authors | genderize_r_ titles | pubmed | wos |
| Gender API | 0.0086 | 0.0289 | 0.034 | 0.04 | 0.1327 | 0.0029 | 0.0123 | 0.0173 | 0.0294 | 0.0853 |
| gender-guesser | 0.0659 | 0.0795 | 0.0787 | 0.1154 | 0.3699 | 0.0031 | 0.0052 | 0.0137 | 0.0105 | 0.0544 |
| genderize.io | 0.0659 | 0.0675 | 0.0809 | 0.0697 | 0.2296 | 0.0091 | 0.0203 | 0.0336 | 0.0235 | 0.0872 |
| NameAPI | 0.063 | 0.0988 | 0.0723 | 0.103 | 0.283 | 0.018 | 0.0158 | 0.0113 | 0.021 | 0.0572 |
| NamSor | 0.043 | 0.0289 | 0.0404 | 0.0697 | 0.214 | 0.006 | 0.0123 | 0.0217 | 0.024 | 0.0741 |

**Table 7** Benchmark 2a: Minimization of inaccuracies constrained to a 5% misclassification error on the entire data set. Displayed are the mean and standard deviation of the values of *errorCoded* for all services, rounded to four decimal figures.

| | Gender API | gender-guesser | genderize.io | NameAPI | NamSor |
|---|---|---|---|---|---|
| mean | 0.0867 | 0.2224 | 0.1454 | 0.1842 | 0.1359 |
| std | 0.0027 | 0.0000 | 0.0010 | 0.0011 | 0.0023 |

than on the others. This is in accordance with the fact that the wos collection incorporates a higher percentage of Asian names, as evidenced by Fig. 1. Regardless, the results follow the general trend: Gender API achieves the smallest fraction of inaccuracies for all data sources, whereas gender-guesser often beats other services in terms of misclassifications. Overall we conclude that the breakdown of errors by data source is consistent with the analysis split by names' origin. Data sets composed of Western names have a much larger chance of being correctly attributed a gender than Asian ones.

## Benchmark 2: Minimization of inaccuracies constrained to a 5% misclassification error

We measure the performance of all services with respect to the fraction of inaccuracies (*errorCoded*) under the constraint that at most 5% of all successfully classified names are misclassified as female or male, i.e., they fulfill *errorCodedWithoutNA* $< 0.05$. This is a realistic constellation for applications requiring the rate of misclassifications not to exceed a given threshold.

### Benchmark 2a: entire data set

We apply our parameter tuning procedure to minimize *errorCoded* on the entire data set and display the averaged test errors per service in Table 7. In each of the 10 runs of 10-fold cross-validation, Gender API produces the lowest error, NamSor the second lowest. In this scenario, it is possible to achieve an average inaccuracy rate under 9% over the whole data set while keeping the misclassification error under 5% with Gender API. NamSor and genderize.io achieve second place with average inaccuracy rates just under 15%.

In order to assess whether the difference in performance is statistically significant, we apply a two-matched-samples *t*-test to the results of the two best services. Since our data set is relatively large and the data sets have been obtained by random sampling, one needs to

**Table 8** Benchmark 2b: minimization of inaccuracies constrained to a 5% misclassification error. Displayed are the values of *errorCoded* for all services and data sources, rounded to four decimal figures.

|  | Gender API | gender-guesser | genderize.io | NameAPI | NamSor |
|---|---|---|---|---|---|
| **zbmath** | 0.0085 | 0.0658 | 0.0687 | 0.1601 | 0.0429 |
| **genderize_r_authors** | 0.029 | 0.0798 | 0.0797 | 0.1037 | 0.0291 |
| **genderize_r_titles** | 0.0339 | 0.0787 | 0.083 | 0.0741 | 0.0402 |
| **pubmed** | 0.04 | 0.1154 | 0.0697 | 0.103 | 0.0697 |
| **wos** | 0.2197 | 0.4478 | 0.3304 | 0.5563 | 0.2758 |

show that the assumption of homogeneity of variance is satisfied prior to applying a *t*-test (see e.g., *Japkowicz & Shah (2014)* (p. 222ff)). Levene's test applied to the errors of Gender API and NamSor yields a large *p*-value of 0.68, and so we conclude that there is (almost) no difference between their variances. The *t*-test returns a very small *p*-value $< 0.01$, while the absolute value of Cohen's d-statistic is 19.1. This means that Gender API's error is statistically significantly lower than NamSor's and the effect size is large.

The parameter tuning procedure on the entire data set for this benchmark leads to optimal parameters close to the default case, maximizing the number of names that are assigned a gender; for instance, the optimal values for Gender API's *accuracy* and *samples* are 57 and 62, respectively, while NamSor's *scale* is tuned to 0.13.

### Benchmark 2b: analysis by data source

For the split analyses we perform one run of 10-fold cross-validation per service and data source; results are displayed in Table 8 and show that Gender API is the best performer in all cases. Incidentally, all services achieve one order of magnitude worse results on the data set wos than on the others. NameAPI and gender-guesser did in fact repeatedly fail to satisfy the constraint, in which case we had to set the error to 1 for the respective fold.

We have applied the Friedman test with the result that the difference in performance among services is statistically significant at significance level 0.01. As a post-hoc test we have applied the Nemenyi test in order to find out which classifiers actually differ. However, the Nemenyi test is not powerful enough to detect statistical significance between any classifiers except Gender API and NameAPI. Since we are particularly interested in the best performers, we have compared Gender API and NamSor using the sign test instead, which counts the number of data sets on which the one classifier outperforms the other. Accordingly, Gender API is significantly better than NamSor at the confidence level 0.05.

## Benchmark 3: Minimization of misclassifications constrained to a 25% non-classification rate

Next we evaluate the effectiveness for achieving correct classifications, i.e., we minimize the rate of misclassifications (*errorCodedWithoutNA*) constrained to the amount of names that cannot be classified being lower than 25% of all assignments (*naCoded* $< 0.25$). This represents a rather frequent situation of wanting to incur as few misclassifications as possible, while at the same time being able to assign a gender to at least a pre-defined fraction of the evaluated names.

Table 9 **Benchmark 3a: Minimization of misclassifications constrained to a 25% non-classification rate on the entire data set.** Displayed are the mean and the standard deviation of *errorCodedWithoutNA* for all services, rounded to four decimal figures.

|  | Gender API | gender-guesser | genderize.io | NameAPI | NamSor |
|---|---|---|---|---|---|
| mean | 0.0088 | 0.0229 | 0.0174 | 0.0302 | 0.0139 |
| std | 0.0015 | 0.0000 | 0.0048 | 0.0009 | 0.0000 |

Table 10 **Benchmark 3b: Minimization of misclassifications constrained to a 25% non-classification rate.** Displayed are the values of *errorCodedWithoutNA* for all services and data sources, rounded to four decimal figures.

|  | Gender API | gender-guesser | genderize.io | NameAPI | NamSor |
|---|---|---|---|---|---|
| zbmath | 0 | 0.0029 | 0.0061 | 0.0105 | 0 |
| genderize_r_authors | 0.0026 | 0.0051 | 0.0144 | 0.0085 | 0.0054 |
| genderize_r_titles | 0.0023 | 0.007 | 0.0173 | 0.0121 | 0.0098 |
| pubmed | 0.0037 | 0.0067 | 0.0037 | 0.0164 | 0.0065 |
| wos | 0.0395 | 1 | 0.0673 | 0.339 | 0.0454 |

### Benchmark 3a: entire data set

As shown in Table 9, Gender API outperforms the other services with a 0.9% misclassification rate, followed by NamSor with 1.4% and genderize.io with 1.7%. To achieve these results, the three services need to be tuned to optimal parameters with higher values of confidence (roughly *accuracy* $> 90$ and *samples* $> 40,000$; *scale* $> 0.70$; *probability* $> 0.95$ and *count* $> 3,500$, respectively).

Since the variances of Gender API and NamSor are not similar, a $t$-test cannot be applied to measure the difference between the two best performers[3]. However, it can be applied to the comparison of Gender API and genderize.io, with the result that Gender API outperforms genderize.io significantly with large effect size.

### Benchmark 3b: analysis by data source

For most of the analyzed data sources Gender API outperforms all other services; error figures are displayed in Table 10. On names from the pubmed collection, Gender API and genderize.io are equally good; on the zbmath subset, Gender API and NamSor achieve a perfect score. Again, all services perform one order of magnitude worse on names from wos than on the other subsets. NameAPI did not satisfy the constraint on various folds, gender-guesser in fact on none of them.

As in Benchmark 2b, the Friedman test shows statistical significance at significance level 0.01. Neither the Nemenyi test nor the sign test confirm significance between the performance of Gender API and NamSor at level 0.05. We conclude that none of the tests considered suitable for comparing Gender API and NamSor are able to confirm that Gender API is statistically significantly better in this case.

### Benchmark 4: Minimization of the weighted error with $w = 0.2$

Finally we analyze the case of minimizing the weighted error with $w = 0.2$, namely the metric that treats all inaccuracies (misclassifications and non-classifications) as errors, but

[3]We have applied the sign test to the test errors in the first trial, since multiple trials violate the independent domain assumption. Gender API outperforms NamSor in seven of 10 folds but this is not significant in terms of the sign test which would require eight folds.

**Table 11   Benchmark 4a: Minimization of the weighted error with weight $w = 0.2$.** Displayed are the mean and the standard deviation of the values of *weightedError* for all services, rounded to four decimal figures.

|       | Gender API | gender-guesser | genderize.io | NameAPI | NamSor |
|-------|------------|----------------|--------------|---------|--------|
| mean  | 0.0458     | 0.0732         | 0.0630       | 0.0674  | 0.0560 |
| std   | 0.0045     | 0.0000         | 0.0017       | 0.0000  | 0.0011 |

**Table 12   Benchmark 4b: Minimization of the weighted error with weight $w = 0.2$.** Displayed are the values of *weightedError* for all services and data sources, rounded to four decimal figures.

|                    | Gender API | gender-guesser | genderize.io | NameAPI | NamSor |
|--------------------|------------|----------------|--------------|---------|--------|
| zbmath             | 0.0039     | 0.0166         | 0.0218       | 0.0293  | 0.0161 |
| genderize_r_authors| 0.013      | 0.0211         | 0.0269       | 0.0337  | 0.013  |
| genderize_r_titles | 0.0176     | 0.0321         | 0.0363       | 0.0251  | 0.022  |
| pubmed             | 0.0243     | 0.0335         | 0.0273       | 0.0386  | 0.0315 |
| wos                | 0.0791     | 0.1407         | 0.1122       | 0.1132  | 0.0986 |

puts five times more weight into the former. This corresponds to an intermediate situation between Benchmarks 1 and 2, and the approach has the flexibility of allowing a continuous range of values for the weight $w$, depending on the particular needs of each analysis.

### Benchmark 4a: entire data set

The best results are achieved by Gender API and NamSor with weighted error values of 0.046 and 0.056, respectively, as shown in Table 11. Since the variance of the two best performers is almost equal, we can apply the $t$-test, which yields statistical significance at significance level 0.01. Also, Cohen's d-statistic confirms that the difference in performance is practically relevant.

As expected, the parameter tuning procedure on the entire data set leads to optimal parameters between those computed for the previous two benchmarks; for instance, the optimal values for Gender API's *accuracy* and *samples* are 75 and 72,003, respectively, while NamSor's *scale* is tuned to 0.41.

### Benchmark 4b: analysis by data source

In Table 12 we present results from minimizing *weightedError* with weight $w = 0.2$ for all services and data sources. Gender API is the best performing service on all data sets; NamSor reaches the second best values on four out of five data sources. The Friedman test shows that the performances are statistically significant at significance level of 0.01. Furthermore, the sign test shows that Gender API is significantly better than NamSor at significance level 0.05.

## DISCUSSION

Name-based gender inference poses numerous challenges. To name a few, the association of a name with gender depends on the cultural and regional context, hence relying on the first name only can be highly error-prone. Transliteration from other alphabets into the

Latin one is known to lead to significant losses of information, thereby excluding entire populations from a reliable classification. Incidentally, the gender of some names might depend not only on culture, but also on historical epoch, and so there exist names that were e.g., typically male in the past and are nowadays female or unisex.

Furthermore, first names are per se embedded into the gender binary, hence this approach reinforces a non-inclusive gender concept and further marginalizes individuals that do not identify as women or men. Clearly, the best way to enrich personal data with this type of demographic information is to ask for self-identification. That would not only increase the correctness of the data; it is also to be preferred under ethical considerations, since it avoids the offensiveness of assigning categories to individuals, while allowing for inclusion of identities beyond the gender binary. Self-identification is not feasible though in large-scale studies of historical data that are typical for bibliometric analyses. Thus the usage of automated methods to infer gender from names or from alternative available details is unavoidable. For a thorough discussion of the ethics of gender identification, see *Matias (2014)* and references therein.

Notwithstanding the above caveats, we have performed a comprehensive comparison of available gender inference tools, testing five services on a manually labeled data set containing 7,076 names, of which 5,779 have a definite gender and are subjected to our analyses. For our evaluations, it would have been desirable to use an open collection of names with labels obtained through self-identification. We are not aware of such a set, thus we have used data based on judgments of third parties. As described in the section 'Assemblage of test data' we have corrected a non-trivial amount of gender assignments, which unfortunately does not preclude potential remaining classification mistakes. Making the test data set public might help to correct them. Furthermore, we have assessed the geographical diversity of our test names, concluding that approximately half of them are of European origin, slightly less than half are Asian, and the remaining 5% are African. Names of persons from the American and Australian continents are considered to descend from these three main regions. We deem this distribution to be appropriate for the task at hand.

We have devised and run various benchmarks to compare all five inference services in several scenarios. In particular, we have computed all performance metrics using the default responses without any further tuning. We have studied the default responses broken down by geographical origin of names and by data source. Additionally, we have performed parameter tuning to search for the optimal values of the confidence indicators that lead to minimization of misclassification or inaccuracy rates, while constraining the maximum error on the other quantity. We have broken down these analyses by data source as well. Finally, we have applied various tests to measure whether the observed differences in performance are statistically significant.

Python package gender-guesser achieves the lowest misclassification rate without parameter tuning for the entire data set, introducing also the smallest gender bias. At the same time it shows poor performance in terms of non-classifications, which is understandable given its comparatively small data base. As the only completely free service with open data and logic, we reckon that it can be useful as a first step of a multi-stage

gender inference procedure. Gender API is the best performer in terms of fraction of inaccuracies, and also in proportion of non-classifications.

When breaking down results without parameter tuning by names' origin we find out that all services perform at least one order of magnitude better on names of European origin than on Asian names. In particular, this translates to poorer results on the wos names subset, which is the less Eurocentric collection of all analyzed data sources. This confirms that assessments of errors in gender inference studies should be made with particular care when the cultural makeup of the analyzed names is unknown. For instance, the genderizeR data subsets employed in the analysis of *Wais (2016)* contain predominantly Western records, which is possibly at the root of the good results produced by a genderize.io service that, as we show, is not particularly well suited for inferring the gender of Asian names. In modern scholarly publications, the share of authors of Asian origin is significant though and thus this caveat needs to be addressed.

Gender API typically achieves the best results after performing parameter tuning to optimize for particular scenarios. It is noteworthy to recall that, in contrast to NamSor and NameAPI, Gender API uses first names only. Using the tuned parameters of Gender API, it is possible to obtain a rate of inaccuracies of 8.7% constrained to not more of 5% of names being misclassified, a result significantly better than that achieved by the second best service NamSor. Likewise, the misclassification error can be made as low as 0.9% while still retaining a classification label for at least 75% of the entire data set. Next in performance is service NamSor, closely followed by genderize.io, both of which achieve a misclassification rate under 2% in that latter scenario. Our results indicate that analyses based on gender predictions by these methods are to be considered as more reliable than regular queries to country censuses or birth name lists.

The addition of further benchmark settings based on supplementary performance metrics might be of interest. For instance, an appropriate measure would be the area under the ROC curve (AUC), which is particularly useful when one of the outcome classes is skewed, as is expected for authorships randomly drawn from databases in STEM disciplines.

A disadvantage of the commercial services though is the lack of transparency regarding their data sources, specifically how records are gathered and processed. Furthermore, the algorithms behind the gender assignments are closed, too, while explanations are usually provided only on the level of technical usage, based on simple examples such as 'John Smith'. Both aspects hamper efforts towards reproducibility of results. At the same time, given the substantial cost of some of the services, a better treatment of specific peculiarities like double names would be expected[4]. To give another example of trivial errors, NameAPI classifies 'paul pinsky' as female with confidence 0.99, while 'Paul Pinsky' or 'Paul pinsky' are returned as male with confidence 0.91. Hence, we recommend potential users to thoroughly test any given service with comprehensive examples before going into production. Our benchmarks and tests aim to provide first solid evidence of the services' capabilities.

For the presented benchmarks we have restricted to names containing at least a first and a last name. Yet the last name may sometimes suffice to infer the gender with high probability; this is e.g., the case for many (though not all) names from Russia or Poland.

[4]Considering the problems arising with middle names, as described for Gender API and NamSor, it might make sense to drop them, or to query with only the first name when the genders of first and middle are in disagreement.

For those services that can handle them, it would be interesting to benchmark on a test set consisting of surnames only. Additionally, our data set contains only names in Latin characters although various services can handle (some) non-Latin alphabets, so it might be desirable to extend the data set in this direction as well. Furthermore, one could expand the study to include more samples of the same first name and test the dependency of gender inference on the last name. Lastly, there exist several more code packages or web services of interest: R package *gender* by Lincoln Mullen (*Mullen, 2016*) utilizes various openly available sources of first names with time range as an additional feature. As explained in *Blevins & Mullen (2015)*, for research of a longer time span, "the changing nature of naming practices" might need to be taken into consideration.

## CONCLUSION

The determination of a person' gender based solely on their name is not straightforward, yet it is a relevant task for plenty of applications, including but not limited to studies of women's representation in tech, media, or academia. In particular, bibliometric analyses of scientific publications in various academic fields have mostly made use of compiled lists of names to label authors of articles as male or female. Less attention has been paid, though, to the quantification of the several errors that one can incur when doing so, or to the advantages of choosing one or another gender assignment method depending on the requirements of the analysis at hand.

Our comparison of five gender inference services in terms of various performance metrics such as inaccuracy, misclassification, and non-classification error rates provides a solid estimation of the accuracy to be expected in name-to-gender inference tasks. Applying these metrics to our data set, which we break down according to the names' geographical origin and data source, we estimate the errors incurred by the services according to the two variables. By performing cross-validated, randomized parameter tuning on a large genderized data set of scientific authors we demonstrate that with three of the surveyed services it is possible to guess the correct gender of more than 98% of names for which a female or male label is returned, while simultaneously leaving less than 25% of records unclassified. Our framework can be trivially extended to account for further gender inference methods.

## ACKNOWLEDGEMENTS

We thank Casper Strømgren for granting us unlimited access to genderize.io and Markus Perl for extending us a discount offer for Gender API. We are indebted to Elian Carsenat for allowing us to freely access NamSor's gender and origin APIs. We are grateful to researchers Vincent Larivière, Cassidy Sugimoto, and Kevin Bonham for sharing their labeled gender data. We thank Marco Tullney for comments on the submitted version. We acknowledge and appreciate the two anonymous referees for carefully reading the manuscript and suggesting substantial improvements.

### Funding

This work was supported by the Grants Programme of the International Council for Science (ICSU), through project "A Global Approach to the Gender Gap in Mathematical, Computing, and Natural Sciences: How to Measure It, How to Reduce It?". The funders had no role in study design, data collection and analysis, decision to publish, or preparation of the manuscript.

### Grant Disclosures

The following grant information was disclosed by the authors:
Grants Programme of the International Council for Science (ICSU).

### Competing Interests

The authors declare there are no competing interests.

### Author Contributions

- Lucía Santamaría conceived and designed the experiments, performed the experiments, analyzed the data, contributed reagents/materials/analysis tools, prepared figures and/or tables, performed the computation work, authored or reviewed drafts of the paper, approved the final draft, prepared the paper for submission.
- Helena Mihaljević conceived and designed the experiments, performed the experiments, analyzed the data, contributed reagents/materials/analysis tools, prepared figures and/or tables, performed the computation work, authored or reviewed drafts of the paper, approved the final draft, set up code and data repository.

### Data Availability

Evaluation of name-based gender inference methods: https://github.com/GenderGapSTEM-PublicationAnalysis/name-gender-inference

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
