# Peer review of "Comparison and benchmark of name-to-gender inference services"

_PeerJ Computer Science, doi:10.7717/peerj-cs.156_

## Round 0.1 · original submission · Minor Revisions

This is a nicely written and useful paper evaluating the relative performance of different techniques / APIs for gender inference from names. It is a good fit for PeerJ CS in both scope and quality.
The Reviewers point out some limitations and possible improvements that I believe can be carried out as minor modification to the existing manuscript. Addressing those minor modifications should be considered a necessary condition for an eventual final acceptance for publication.
Some comments about the limitations of previously published work should be more carefully phrased, as they do not appear fully supported neither by the actual claims of the cited work, nor by the evidence presented by the Authors. This applies in particular to the comments about the cited work by Karimi et al.

Reviewer 1 ·

Basic reporting

This paper describes an evaluation of available methods (mostly web APIs) that allow to infer the gender of a person based on its name.
The paper is overall written well and can be easily understood. Related work is described sufficiently (although given my background I cannot guarantee completeness). The paper structure is adequate. Raw data and source code for the paper is available and makes the impression to be complete.

Personally, I find it somewhat strange to cite the address of the raw data of a paper as a regular citation.

Experimental design

The paper pursues a clear and well defined goal. The outcome of the paper can be helpful to researchers in various academic disciplines.

The paper is currently missing a statistical analysis of the findings, i.e., it is unclear which differences are statistically significant. I suggest to add the respective tests, see e.g., "Japkowicz, Nathalie, and Mohak Shah. Evaluating learning algorithms: a classification perspective. Cambridge University Press, 2011."

Validity of the findings

The discussion should go into more detail with respect to bias potentially be implied by data used for the study (e.g., what are the supposed origins of the names, what is the share of Asian/African names, etc...). Looking at the raw data, I have a loose assumption that parts of the dataset (zbmath) are focused on western names. Indeed, it would be also interesting to split the analysis into the different data sources and report results. That would give the reader an impression how strongly results vary for different (sub-) datasets.

The setting of Benchmark 2 is not completely clear to me. Is it "at most 5% of all names" (as in the paper, line 400) or "at most 5% of all classified names"? I would like this Benchmark to be described in more detail.

For the setting in Benchmark 3, reporting the RoC curve would be the appropriate measure to use in my opinion. I strongly suggest to add that to the paper. See: "Bradley, Andrew P. "The use of the area under the ROC curve in the evaluation of machine learning algorithms." Pattern recognition 30.7 (1997): 1145-1159."

Table 4, Table 5, and Table 6 contain very little information. I would suggest to just report mean _and standard deviation_ of the iterations.

Additional comments

Figure 1 and its meaning/relevance should be described in more detail.

Tables use German instead of English floating point separators (i.e. "0,002" instead of "0.002")

Reviewer 2 ·

Basic reporting

This paper represents a very interesting and worthy contribution to the field of automatic gender inference. Written English quality is consistent and of high quality, and both the paper and the overall argument are well-structured. Its problematization within the context of academic publishing and bibliometrics is both extensive and relatively up-to-date. However, all publications cited in the references section strictly pertain to automatic gender inference; given the importance and sophistication of the performance evaluation metrics and procedures used in this research, reference to seminal works pertaining to predictive analytics and confusion matrices would have been most welcome (at least some of them must have been consulted in doing this research). Finally, both analysis and discussions are directly related to the objective and problematics of the paper.

Experimental design

This article fits nicely within the aims and scope of this journal. The investigation procedure is well described and could easily be replicated. The contribution of this research to the existing body of knowledge is also well stated. In sum, nothing negative to report regarding experimental design.

Validity of the findings

Interpretation of the results is prudent, yet unambiguous. The authors have done a nice work collecting, compiling, and cleaning all the data necessary for the project. Two points need to be emphasized, however. In the data collection phase, the authors used nltk's Named Entity Recognizer to identify names in bibliographic records drawn from WoS. We know from experience that this algorithm has non-negligible efficiency and accuracy problems, especially in comparison to other, freely available alternatives such as Stanford's CoreNLP package. A few comments thereon would be desirable. Also, while the authors do mention the time- and country-dependency of automatic gender detection based on first names, the impact of these characteristics on the performance of the algorithms they evaluate as well as on the significance and scope of their research seems somewhat downplayed. To give one example (surely known by the authors), China represents the second biggest country in the world in terms of research output, but existing gender detection algorithms based on first names often perform poorly when it comes to disambiguating the gender of researchers of Asian origin. The fact that this isn't addressed in the evaluation procedure certainly affects the scope and impact of the research results, and the absence of any thorough discussion thereof reflects somewhat badly on the paper.

Additional comments

No comments.

---

## Round 0.2 · accepted · Accept

The revised manuscript satisfactorily addresses all of the minor points raised by the Reviewers, hence it is now suitable for publication.